# TCAD-Based Investigation of a 650 V 4H-SiC Trench MOSFET with a Hetero-Junction Body Diode

**DOI:** 10.3390/mi13101741

**Published:** 2022-10-14

**Authors:** Ruoyu Wang, Jingwei Guo, Chang Liu, Hao Wu, Zhiyong Huang, Shengdong Hu

**Affiliations:** 1Chongqing Engineering Laboratory of High Performance Integrated Circuits, School of Microelectronics and Communication Engineering, Chongqing University, Chongqing 400044, China; 2The National Laboratory of Science and Technology on Analog Integrated Circuits, Chongqing 401332, China

**Keywords:** silicon carbide, heterojunction diode, trench MOSFET

## Abstract

In this paper, a 650 V 4H-SiC trench Metal-Oxide-Semiconductor Field-Effect Transistor (MOSFET) with a hetero-junction diode (HJD) and double current spreading layers (CSLs) is proposed and studied based on Sentaurus TCAD simulation. The HJD suppresses the turn-on of the parasitic body diode and improves the performance in the third quadrant. CSLs with different doping concentrations help to lower the on-state resistance as well as the gate-drain capacitance. As a result, the on-state resistance is decreased by 47.82% while the breakdown voltage remains the same and the turn-on and turn-off losses of the proposed structure are reduced by 83.39% and 68.18% respectively, compared to the conventional structure.

## 1. Introduction

Silicon carbide (SiC), as known as wide band-gap semiconductors, is widely used in power devices such as MOSFET, providing lower on-state resistance (*R*_on_), higher breakdown voltage (BV), and better frequency characteristics [1,2,3,4]. However, the body diode of SiC MOSFETs has a higher turn-on voltage compared with Si MOSFETs, leading to higher switching loss and causing the inherent bipolar degradation effect [5]. Moreover, due to the basal plane dislocations (BPDs), the long-term reliability of the device is a concern [6].

To solve the problem of the body diode, many works of literature have been published. Schottky Barrier Diodes (SBDs) are widely used to replace the body diode working as reverse diodes in SiC MOSFETS, such as the SiC MOSFET with the merged junction barrier controlled Schottky rectifier, the SiC MOSFET with the built-in SBD and the SiC split-gate MOSFET with merged SBD [7,8,9]. While SBDs can optimize the recovering characteristics, they can consume a large part of the active chip area, and lead to high leakage current.

A SiC MOSFET with a MOS-Channel diode is another solution raised recently, but a strong electric field at the edge of the dummy gate may cause reliability issues [10,11,12].

Now heterojunction diode (HJD) has become a new option. HJD formed between poly-silicon and 4H-SiC presents similar characteristics as SBD [13], but needs less area for chips. Several structures have been proposed [14,15,16]. A split-gate SiC trench MOSFET with a P-poly/SiC hetero-junction diode has been proposed for optimized reverse recovery characteristics and low switching loss [17]. Furthermore, SiC MOSFET with integrated n-/n-type poly-Si/SiC heterojunction freewheeling diode has been proposed, offering a lower *V*_f_, but at the cost of BV [18].

In this paper, a new 650 V SiC trench MOSFET with an embedded heterojunction diode is proposed and studied with numerical simulation. The novel structure is compared with the conventional asymmetric channel SiC trench MOSFET (C-MOS). Simulation has shown that the conventional structure has excellent temperature stability and gate oxide reliability but did not concern the problem of the body diode; therefore, a heterojunction diode is embedded into the proposed structure, which prominently improves the recovering characteristics and switching loss. Furthermore, to improve the on-state performance, CSLs attached with a shallow trench gate are introduced as well [19].

## 2. Device Structure

Figure 1a,b show the cross-sectional views of the C-MOS and the proposed structure. Compared with the C-MOS, the proposed structure has a P+ doped poly-Si, a deep P+ well surrounding the p-base, and a shallow poly-silicon gate trench as well as CSLs of different doping concentrations. The P+ poly-Si is connected to the source metal, forming an HJD with n-4H-SiC.

The p-poly and the CSL2 form the HJD. Figure 2 shows the energy band diagram of P-poly/N-SiC heterojunction diode. The thickness of the poly-Si is 0.12 μm horizontally and 0.06 μm vertically. The doping concentration is 1 × 10^20^ cm^−3^. The energy gaps in the HJD for conduction and valence bands are 0.44 eV and 1.82 eV, respectively, providing unipolar action like SBD. The barrier height *Φ*_BN_ is about 1.48 eV. When the forward bias was applied, the built-in potential began decreasing, and the conduction band of the CSL2 rises, causing the electrons in the CSL2 to easily migrate to the P-poly region, but the high energy barrier prevents the holes from passing from CSL2 to the P-poly, providing a low forward voltage (*V*_f_) while keeping the breakdown voltage high.

The gate depths of the C-MOS and the HJD-TMOS are 0.62 μm and 1.2 μm respectively. Since the shallow trench gate enlarges the Junction Field-Effect Transistor (JFET) area and reduces the peak electric field and the *C*_rss_, an over-short channel length will lead to high leakage current and even breakdown of the BV. So, the shallow trench gate of the HJD-MOS is set to 0.62 μm. The thicknesses of the gate oxide layer in the wall and bottom are 60 and 120 nm, respectively.

The deep P+ well surrounding the p-base was set to deplete the CSL, providing a strong pinch-off effect to lower the leaking current and guarantee the BV. The C-MOS and the proposed HJD-TMOS have the same doping concentration and closed device dimensions. The depth of the P+ well is 1.4 μm, and the depth of the P-base is 0.8 μm. The depth of the CSL1 is the same as the deep P+ well, and its thickness is 0.2 μm. The doping concentration of the CSL1 is 5 × 10^17^ cm^−3^. The depth of the CSL2 is 1.6 μm and its doping concentration is 2 × 10^16^ cm^−3^. The other parameters of the two structures are summarized in Table 1.

For device simulations, Sentaurus TCAD (Synopsys Inc., CA, USA) is used to reveal the electric characteristics. SRH, AUGER, and OkutoCrowwell are used as models to describe trap-assisted recombination [20,21,22], the non-radiative process involving three carriers, and the breakdown analysis, respectively. The temperature is set to 300 K.

## 3. Simulation Results and Discussions

Figure 3 shows the influence of different widths of CSL1 (*W*_c1_) on BV, on-state resistance, and switching loss of the structure. When the *W*_c1_ rises, the BV and Ron both decrease, but the switching loss of the structure decreases first, reaches the bottom when *W*_c1_ is 200 nm, and rises afterward. When *W*_c1_ is less than 200 nm, the majority carriers of the JFET area are holes, and with the increase in *W*_c1_, the concentration of electrons also rises. When *W*_c1_ reaches 200 nm, the JFET area is depleted, and the *Q*_gd_ reaches its lowest value, minimizing the switching loss value as well. When *W*_c1_ rises, the majority carriers change into electrons, and the switching loss rises again. Moreover, the decrease in the *R*_on_ is gradually smooth when *W*_c1_ increases to 200 nm, since a further increase in *W*_c1_ does not help to widen the current path. Therefore, the *W*_c1_ is set as 200 nm in the structure.

Figure 4 shows the electric field distributions of C-MOS and the proposed structure in the off state when the *V*_DS_ is 650 V. According to research [23], the maximum oxide electric field (*E*_MOX_) to obtain a lifetime of more than 10 years was estimated to be 2.7 MV·cm^−1^ in blocking state. Because of the deep P+ shielding layer, both of the structures have a lower electric field than 2.7 MV·cm^−1^ at the gate oxide. Since the proposed structure has shallower gate oxide, when the source is biased zero, and the drain is applied with large bias voltage, the P+ shielding region can deplete the CSLs, preventing the high drain electric field from affecting the trench gate, so the peak electric field around the trench gate corner is also lower than the C-MOS, which means it has higher gate reliability. As in the corner of the P+ well, the two structures have similar peak electric fields.

Figure 5 shows the first quadrant I-V characteristics of the C-MOS and the proposed structure. The voltage of the gate (*V*_gs_) is 15 V. The *R*_on, sp_ of the proposed structure is 47.82% lower than the C-MOS. The CSLs provide a better depletion area at the JFET region. Figure 5b,c show the current path of the two structures, the width of the proposed structure is wider than the C-MOS, making a lower on-state resistance. At the same time, the saturation current of the proposed structure is lower than the C-MOS, providing a better short-circuit capability.

Figure 6 shows the reverse characteristics when *V*_gs_ is set to −5 V. The turn-on voltage (*V*_f_) of the HJD is 1.2 V compared to 2.8 V of the body diode, leading to a remarkable reduction in dead-time loss.

As shown in Figure 7a,b, in the proposed structure, the reverse current goes through HJD rather than through body diode as the C-MOS, which avoids the inherent bipolar degradation effect and the BPDs.

Figure 8 compares the off-state characteristics of the two structures. The BV of the proposed structure and the C-MOS are 1084 V and 1043 V respectively. The doping concentration of the CSL is much lower than the deep P+ regions, making it easily depleted at high drain voltage. The shallow gate trench leads to a higher JFET resistance, creating a balance between *R*_on_ and BV. Although the wider current path may lead to a higher leak current in off-state conditions, the strong pinch-off effect provided by deep P+ well helps to offset the problem.

Figure 9 shows the gate charges of the two devices. The proposed structure has a much lower miller plateau than the C-MOS because the distance between the gate and source electrodes is greatly reduced, leading to lower *C*_gs_. The deep P+ region of the proposed structure provides a capacitive shielding effect, helping to reduce the *C*_rss_. The *C*_rss_ can be expressed by the equation [24]:(1)$$C_{\mathrm{rss}}=\frac{C_{\mathrm{OX}}C_{\mathrm{dep}}}{C_{\mathrm{OX}}+C_{\mathrm{dep}}}$$
in which *C*_OX_ is the oxide capacitance and *C*_dep_ is the bulk depletion capacitance. The shallow gate provides a low *C*_OX_, since the active gate area is reduced. As a result, the *Q*_gd_ is reduced by 82.78% in total.

The switching characteristics of the two structures are shown in Figure 10. The test was applied with the typical testing circuit with the inductive load shown in Figure 10a. However, the reverse recovery current spike of the proposed structure is larger than the C-MOS because of the mismatch between the gate resistance and the decreased *C*_gd_. The body diode of the C-MOS works as a bipolar device, while the HJD in the proposed structure works as a unipolar device. When the diode is turned off, the number of minority carriers in the proposed structure is much less than the C-MOS, leading to a smaller reverse recovery time (t_rr_) and the reverse recovery charge (*Q*_rr_). Moreover, the reduced *C*_gd_ and *C*_gs_ brought by the shallow gate make further efforts to decrease the turn-on and turn-off time. As a result, the turn-on loss and turn-off loss decreased by 83.39% and 68.18% respectively, and the total switching loss is reduced by 77.78%, as can be seen in Figure 11.

Table 2 Summarizes the electrical characteristics of the C-MOS and the proposed structure.

## 4. Proposed Fabrication Process

Considering the feasibility of the proposed structure, the fabrication process is shown in Figure 12. The main process of the proposed structure is the same as the conventional structure but needs more steps to form the CSLs and the P+ poly-silicon. First, the N drift and the CSL1 region are formed by epitaxial growth on the N+ substrate. Then, the CSL2 and the deep P+ well are formed by ion implantation [25]. Then, on the top of the P+ well on the right, through double implantation [26], the P- base and the N+ source region are formed. After that, the HJD region and gate region are trenched at the same time. Then the gate is formed with thermal oxidation and poly-silicon deposition. HJD is then formed by depositing P+-doped poly-silicon. Lastly, the metal layer is connected to the source.

## 5. Conclusions

In this paper, a novel 650 V 4H-SiC trench MOSFET is proposed and studied by Sentaurus TCAD simulation which features a heterojunction diode and CSLs with different doping concentrations. The HJD helps to improve the third quadrant characteristics and recovery performance, while the CSLs provide a lower on-state resistance. Compared to C-MOS, the *R*_on, sp_ is decreased by 47.82%, the *Q*_gd_ decreased by 82.78, and the switching loss is reduced by 77.78%, indicating a superior candidate in high-frequency situations.

## Figures and Tables

**Figure 1 micromachines-13-01741-f001:**
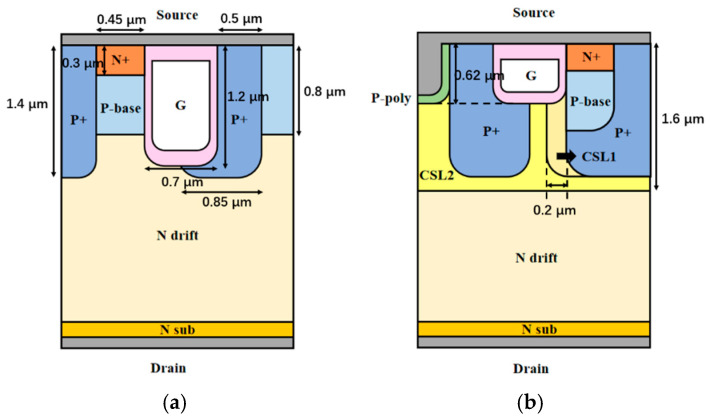
Cross-sectional views of (**a**) C-MOS and (**b**) proposed structure.

**Figure 2 micromachines-13-01741-f002:**
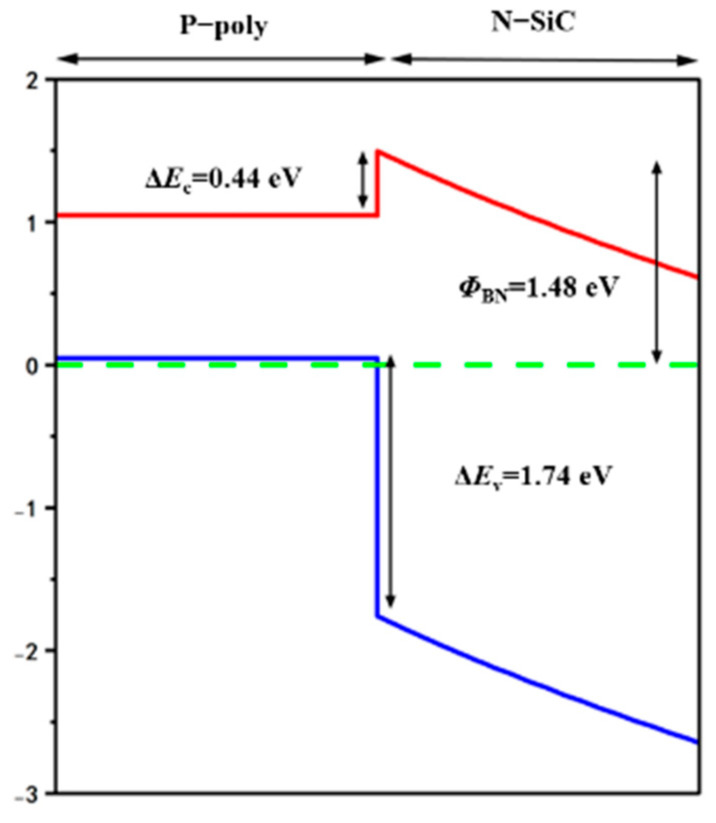
Band diagram of P-poly/N-SiC heterojunction diode.

**Figure 3 micromachines-13-01741-f003:**
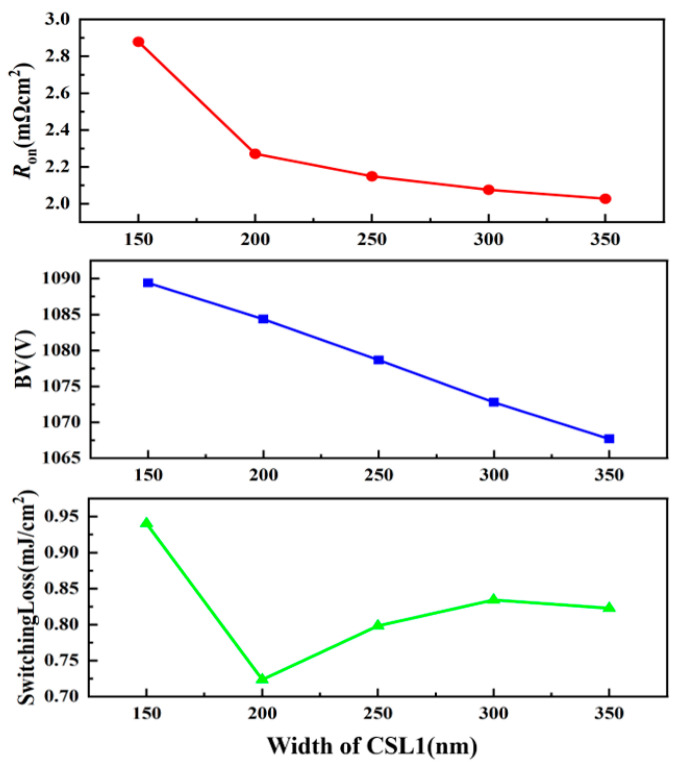
The influence of the width of CSL1 (*W*_c1_) on the Ron (*V*_GS_ = 15 V), BV (*V*_GS_ = 0 V) and switching loss.

**Figure 4 micromachines-13-01741-f004:**
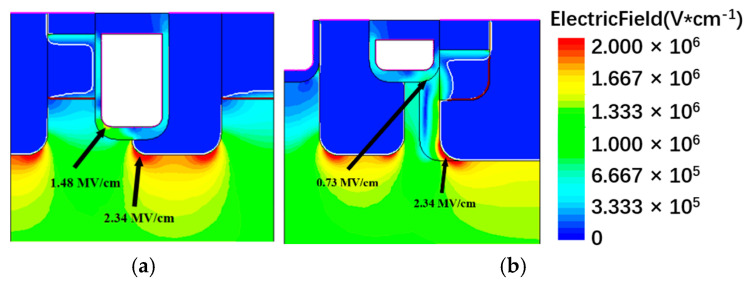
The electric field distribution of (**a**) C-MOS, (**b**) proposed structure.

**Figure 5 micromachines-13-01741-f005:**
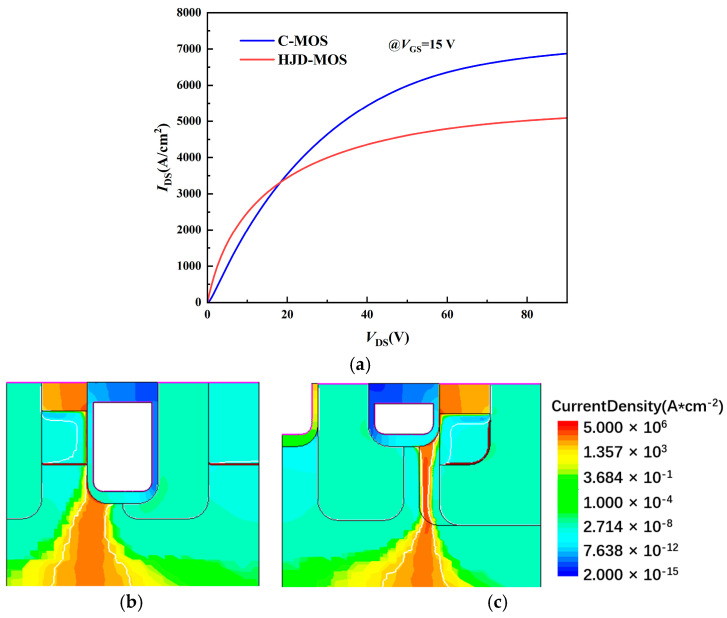
The (**a**) first quadrant I-V characteristics of the C-MOS and the proposed structure and the current path of (**b**) C-MOS and (**c**) proposed structure.

**Figure 6 micromachines-13-01741-f006:**
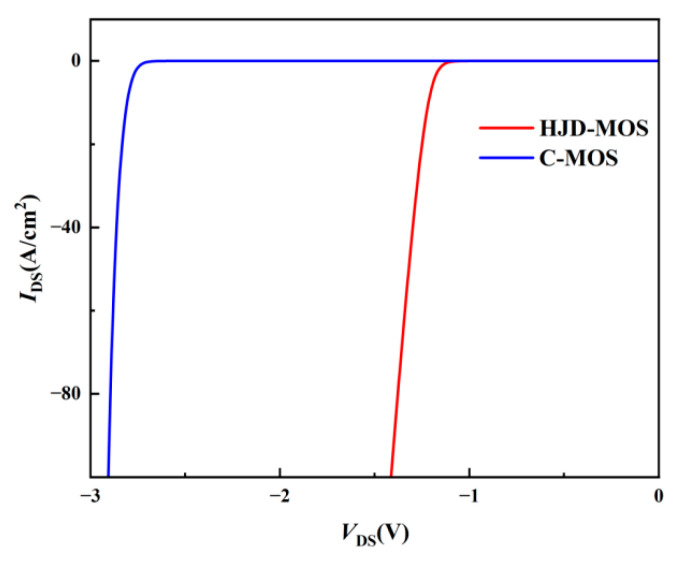
The reverse characters of the C-MOS and the proposed structure.

**Figure 7 micromachines-13-01741-f007:**
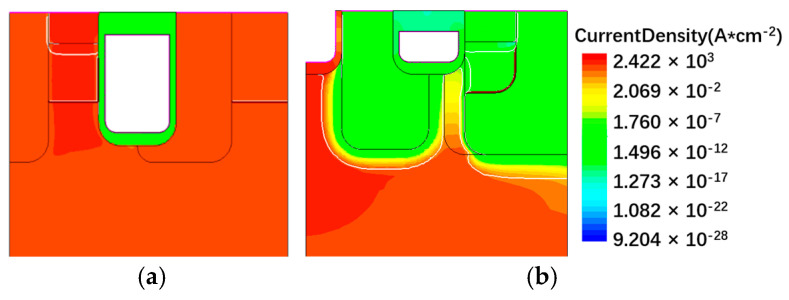
The current distribution of (**a**) C-MOS and (**b**) proposed structure.

**Figure 8 micromachines-13-01741-f008:**
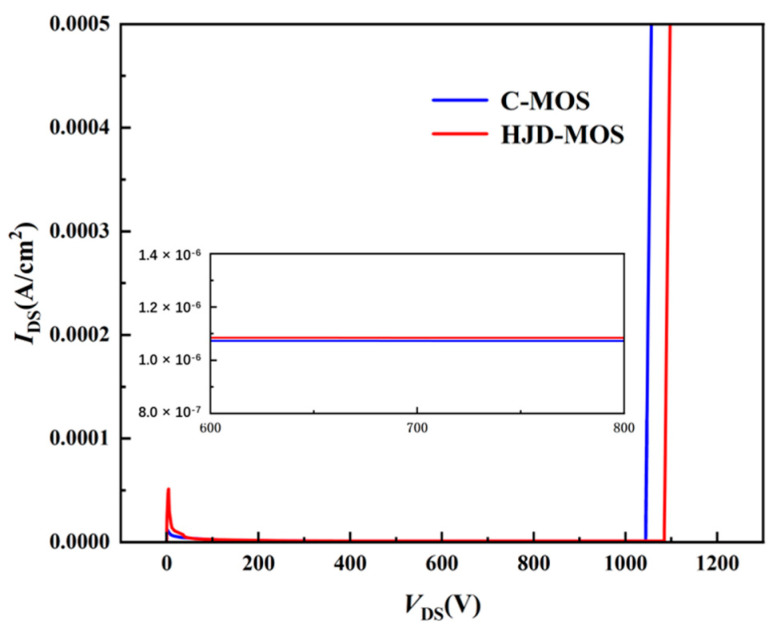
The off-state characteristics of the C-MOS and the proposed structure.

**Figure 9 micromachines-13-01741-f009:**
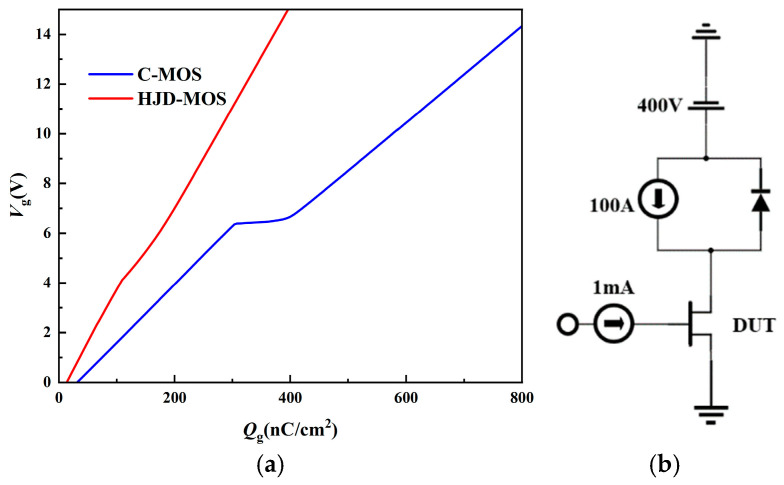
(**a**) The gate charge of the C-MOS and the proposed structure. (**b**) The test circuit of the gate charge.

**Figure 10 micromachines-13-01741-f010:**
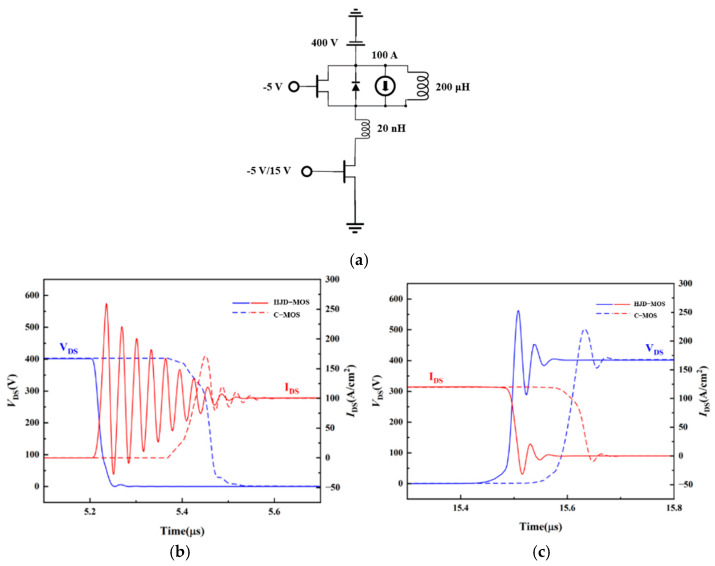
(**a**) Double pulse test circuits. (**b**) Turn-on and (**c**) turn-off waveform of the C-MOS and the proposed structure.

**Figure 11 micromachines-13-01741-f011:**
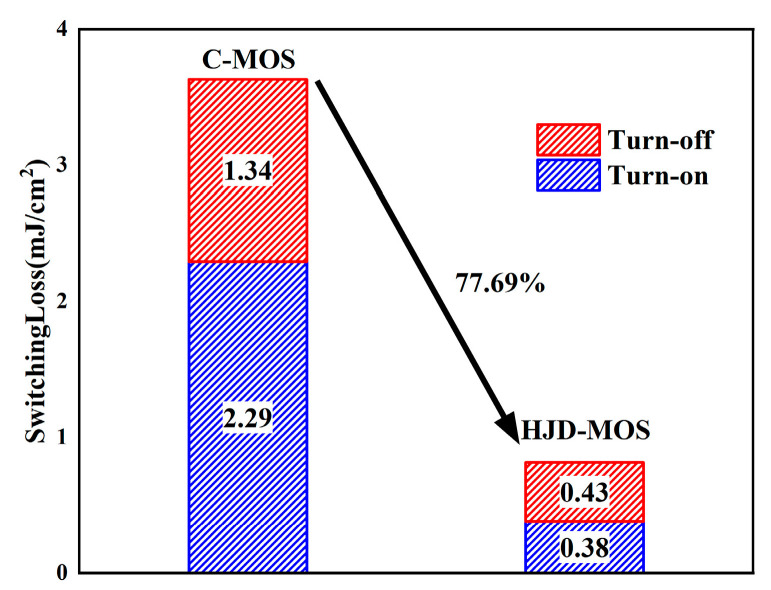
The gate charge of the C-MOS and the proposed structure.

**Figure 12 micromachines-13-01741-f012:**
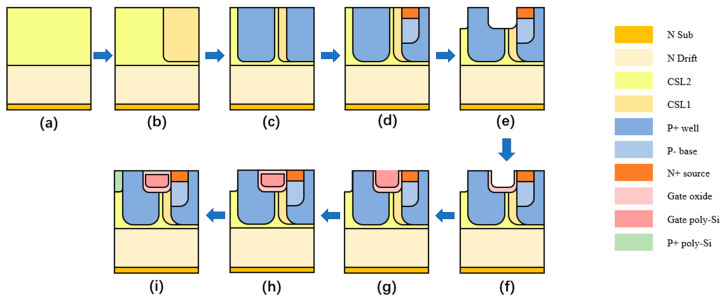
The proposed fabrication process. (**a**) Epitaxial N- drift and CSL1 (**b**) Forming of CSL2 by ion implantation (**c**) Forming of P+ well by ion implantation (**d**) Forming of P- base and N+ source by double ion implantation (**e**) Annealing, trench gate region and HJD region (**f**) Thermal oxidation to form gate oxide (**g**) Depositing of gate poly-silicon (**h**) Depositing of gate oxide (**i**) Depositing of P+-doped poly-silicon of HJD region.

**Table 1 micromachines-13-01741-t001:** Parameters of C-MOS and Proposed HJD-MOS.

Parameters	C-MOS	HJD-MOS	Unit
Cell pitch	2.5	2.5	μm
Depth of P+ well	1.4	1.4	μm
Depth of P-base	0.8	0.8	μm
Depth of N+ source	0.3	0.3	μm
Depth of trench gate	1.2	0.62	μm
Depth of CSL 1	/	1.35	μm
Depth of CSL 2	/	1.6	μm
Width of P+ well (left/up)	0.35	0.5	μm
Width of P+ well (left/down)	0.35	0.85	μm
Width of P+ well (right/up)	0.5	0.45	μm
Width of P+ well (right/down)	0.85	0.95	μm
Width of trench gate	0.7	0.7	μm
Width of N+ source	0.45	0.45	μm
Width of P-base	0.45	0.45	μm
Width of CSL 1	/	0.2	μm
Width of P-poly	/	0.35	μm
Thickness of gate oxide (bottom)	120	120	nm
Thickness of P-poly (side)	/	60	nm
Thickness of gate oxide (up)	200	200	nm
Thickness of gate oxide (side)	60	60	nm
Thickness of P-poly (bottom)	/	120	nm
P+ well doping	1 × 10^19^	1 × 10^19^	cm^−3^
N+ source doping	1 × 10^19^	1 × 10^19^	cm^−3^
P-base doping	3 × 10^17^	3 × 10^17^	cm^−3^
N drift doping	7.5 × 10^15^	7.5 × 10^15^	cm^−3^
N sub doping	1 × 10^19^	1 × 10^19^	cm^−3^
CSL 1 doping	/	5 × 10^17^	cm^−3^
CSL 2 doping	/	2 × 10^16^	cm^−3^
P-poly doping	/	1 × 10^20^	cm^−3^

**Table 2 micromachines-13-01741-t002:** Device Characteristics.

Symbol	C-MOS	HJD-MOS	Unit
BV	1043	1084	V
*R* _on_	6.51	2.27	mΩ×cm^2^
*Q* _gd_	84	14	nC/cm^2^
*V* _f_	2.8	1.2	V
BFOM ^a^	167	518	MW/cm^2^
*R*_on, sp_ × *Q*_gd_	546.84	31.78	mΩ×nC

^a^ BFOM = BV^2^/*R*_on, sp_.

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
