# Peer review of "TCAD-Based Investigation of a 650 V 4H-SiC Trench MOSFET with a Hetero-Junction Body Diode"

_micromachines, 2022, doi:10.3390/mi13101741_

Round 1

Reviewer 1 Report (Previous Reviewer 2)

The aim, methodology and results of this TCAD investigations are well structured and presented. I only suggest a minor language revision.

Author Response

Dear Reviewer:

Thank you for your letter and for the reviewers’ comments concerning our manuscript entitled “TCAD-Based Investigation of a 650 V 4H-SiC Trench MOSFET with a Hetero-Junction Body Diode” (micromachines-1949531).

Those comments are all valuable and helpful for revising and improving our paper, as well as the important guiding significance to our researches. We have studied comments carefully and have made correction which we hope to meet with approval.

Reviewer 2 Report (New Reviewer)

R. Wang et al. have proposed a 650 V 4H-SiC trench MOSFET with a hetero-junction diode and double current spreading layers through a simulation methodology. The computational investigation has been conducted via the Sentaurus TCAD simulation. Some interesting results have been emphasized. I believe that the treatment of the following comments can strengthen the manuscript and polish its contribution increasingly. The paper is well organized and informative. The reviewer's comments are below:

1) The most of interpretations dealing with the recorded results, are superficial. I recommend dealing with the underlying physics

2) Fig. 3 lacks the title and unity of abscissa coordinate. 

3) I recommend inserting a comparison table benchmarking the proposed structure with those available in the literature in terms of the estimated figure of merit.

4) For more clarity, the dimensional parameters corresponding to the indicated parameters in Tab. 1, should be inserted in Fig. 1(a) and (b). 

5) The inset in Fig. 9 is not really clear. I recommend enhancing it. 

6) It is more appropriate if the bottom curve in Fig. 8 will be inserted as inset in the top figure.

Good Luck.

Author Response

Dear Reviewer:

Thank you for your letter and for the reviewers’ comments concerning our manuscript entitled “TCAD-Based Investigation of a 650 V 4H-SiC Trench MOSFET with a Hetero-Junction Body Diode” (micromachines-1949531).

Those comments are all valuable and helpful for revising and improving our paper, as well as the important guiding significance to our researches. We have studied comments carefully and have made correction which we hope to meet with approval.

Round 2

Reviewer 2 Report (New Reviewer)

The paper is well revised.

This manuscript is a resubmission of an earlier submission. The following is a list of the peer review reports and author responses from that submission.

Round 1

Reviewer 1 Report

This is purely a device simulation study.  No experiments or fabrication of device structure has been carried out.  There is no novelty in the structure.  All the elements such as polysilicon/SiC hetrojunction diode, current spreading layer etc. are well known.  This paper should not be published unless the structure can be fabricated.  No suggestions have been made as to how to implement certain features during actual fabrication.

Reviewer 2 Report

This study presents an improved cell layout for trench SiC FETs. The content, structure, and presentation are adequate. I only have two minor comments: 

1 - Commenting Fig. 6 you mention that the 3rd quadrant characteristic "shows a linear relationship". This is not true. The characteristic is still exponential, although the knee happens at a lower magnitude on the Vds axis.

2 - The ringing and current spike in Fig 10a are actually quite significant, could you comment a bit further as for the reason this happens and how to possibly mitigate it?